# Potential Role of the Glycogen-Targeting Phosphatase Regulatory Subunit in Airway Hyperresponsiveness in Asthma

**DOI:** 10.3390/biomedicines13123111

**Published:** 2025-12-17

**Authors:** Marisol Alvarez-González, Elizabeth Eslava-De Jesús, Blanca Bazan-Perkins

**Affiliations:** 1Laboratorio de Inmunofarmacología, Instituto Nacional de Enfermedades Respiratorias Ismael Cosio Villegas, Calzada de Tlalpan 4502, Colonia Sección XVI, Mexico City 14080, Mexico; elizabeth_edj@ciencias.unam.mx; 2Escuela de Medicina y Ciencias de la Salud, Tecnologico de Monterrey, Mexico City 14380, Mexico

**Keywords:** airway smooth muscle, RG1, MYPT1, ROCK2, airway hyperresponsiveness, asthma model

## Abstract

**Objective**: Airway hyperresponsiveness (AHR) is a hallmark feature of asthma; however, its precise molecular mechanisms remain incompletely defined. In this study, we investigated protein expression in airway smooth muscle that may contribute to AHR, using an experimental model of ovalbumin-induced allergic asthma. **Methods**: Guinea pigs were sensitized and challenged with ovalbumin. Airway responsiveness to histamine was assessed, and proteomic analysis of the tracheal tissue was conducted using electrophoresis followed by MALDI/TOF-TOF mass spectrometry. Specific protein bands corresponding to the myosin phosphatase target subunit 1 (MYPT1) were analyzed, and regulatory subunit of glycogen-targeted phosphatase 1 (RG1) was further evaluated through immunohistochemistry. **Results**: MYPT1, previously associated with AHR, was not detected in the proteomic analysis. Interestingly, an RG1 peptide was identified. Immunohistochemistry showed a differential expression pattern was observed for the RG1 and Rho-associated protein kinase 2 (ROCK2), both of which were significantly upregulated in airway smooth muscle and positively correlated with the degree of AHR. Moreover, a significant positive correlation was observed between RG1 and ROCK2 expression levels. MYPT1 and its phosphorylated forms (Thr^696^ and Thr^850^), along with ROCK1 immunostaining, did not differ from controls. **Conclusions**: These findings suggest that RG1, along with ROCK2, may play an important role in airway hyperresponsiveness characteristic of asthma.

## 1. Introduction

Asthma is a chronic and heterogeneous disease with high global prevalence, particularly in its allergic form. It is characterized by persistent airway inflammation and airway hyperresponsiveness (AHR) [1,2], defined as an exaggerated contraction of the bronchial smooth muscle in response to various stimuli. Understanding the mechanisms underlying smooth muscle contraction and the hypercontractility observed in AHR has been challenging due to the involvement of multiple signaling pathways. Nevertheless, several studies have identified the dysregulation of key proteins as a determining factor in modulating the asthmatic response [3].

Smooth muscle contraction depends on the interaction between actin and myosin filaments. Myosin consists of heavy (MHC) and light (MLC) chains, with MLC17 and MLC20 having approximate molecular weights of 17 kDa and 20 kDa, respectively. Although the precise role of MLC17 remains unclear, MLC20 is known to be phosphorylated by myosin light chain kinase (MLCK) at the serine 19 residue in response to increased intracellular calcium levels. This phosphorylation induces a conformational change in myosin, enhances its ATPase activity, and facilitates actomyosin interaction, thereby promoting smooth muscle contraction [4].

In contrast to the function of MLCK, myosin light chain phosphatase (MLCP) promotes smooth muscle relaxation through the dephosphorylation of MLC20. This holoenzyme consists of three subunits: a catalytic subunit (PP1c), a small subunit (M20), and a regulatory subunit, the myosin phosphatase target subunit 1 (MYPT1) [3,5]. The activity and subcellular localization of PP1c depend on its interaction with distinct regulatory subunits. While MYPT1 directs the holoenzyme to actin and myosin filaments [6], other subunits, such as RG1, target PP1c to complexes involved in glycogen metabolism [7,8].

MYPT1 is a large regulatory protein (~1030 amino acids) with an estimated molecular weight ranging from 115 to 130–133 kDa, depending on the specific type of smooth muscle in which it is expressed. This variability is attributed to the presence of isoforms generated by alternative splicing, which modulate its structural and regulatory properties [9]. Moreover, proteolytic fragments of MYPT1—derived from the N-terminal region—have been experimentally described, with approximate molecular weights of 70–75 kDa and 50–58 kDa. These fragments retain the ability to associate with PP1c. Unlike the full-length 130–133 kDa form, these truncated fragments have been associated with increased MLC20 phosphorylation and enhanced smooth muscle contraction [10,11,12].

Multiple studies utilizing asthma models and human airway smooth muscle (ASM) have indicated that dysregulation of MYPT1 contributes to the hypercontractile state characteristic of AHR, identifying MYPT1 as a crucial regulator of the bronchoconstrictive response. This negative regulation is linked to elevated levels of phosphorylated myosin light chain (MLC), which increases smooth muscle contractility and worsens bronchoconstriction [3,13,14]. However, despite its functional importance, the molecular pathways connecting smooth muscle dysfunction to AHR are not fully understood [15,16,17]. To date, neither the presence nor the functional role of proteolytic MYPT1 fragments have been described in ASM. Considering that such fragments modulate PP1c interaction and enhance contractility in other systems, it is plausible that MYPT1 may exist in distinct proteolytic forms with varying molecular weights that could contribute to the amplification of AHR. This study aimed to examine the expression of MYPT1 and its potential protein fragments and to profile the protein expression associated with AHR in ASM, using an experimental model of allergic asthma in ovalbumin (OVA)-sensitized and challenged guinea pigs.

## 2. Materials and Methods

### 2.1. Animals and Experimental Conditions

Male outbred guinea pigs of the HsdPoc strain, weighing between 0.35 and 0.40 kg, were procured from Harlan in Mexico City, Mexico. Only young males were utilized, as during the model standardization phase, young females (approximately 450 g) consistently failed to develop AHR—a critical requirement for establishing the asthma model (unpublished data). This adjustment to the original protocol was made to minimize biological variability and ensure reproducible airway responses across experiments. The guinea pigs were housed under controlled laboratory conditions, maintained at 50–70% humidity, a temperature of 21 ± 1 °C, with 12-h light/dark cycles, filtered air, and sterilized bedding. All procedures adhered to the Mexican Official Standards NOM-062-ZOO-1999 and NOM-087-ECOL-SSA1-2002 [18,19]. Animal procedures were conducted following protocols approved in 2016 by the Scientific and Bioethics Committee of the Instituto Nacional de Enfermedades Respiratorias Ismael Cosío Villegas (Approval number: B2416; IRB organization: IORG0003948).

### 2.2. Allergic Asthma Model and Barometric Plethysmography

The number of animals used was determined by adhering to the minimum required to follow the principles of the 3Rs (Replacement, Reduction, and Refinement), backed by previous studies that have demonstrated the reliability and robustness of this asthma model. Randomization was performed by assigning guinea pigs to the control or asthma model groups randomly, ensuring unbiased allocation. Animals were individually identified with temporary, non-invasive ink markings applied to the fur, allowing continuous identification without affecting welfare. The allergic asthma model was developed based on the methodology of Campos et al. [20], with modifications to the protocol that included adjusting the OVA doses during sensitization and adding additional nebulized challenges [21]. In accordance with ethical guidelines, guinea pigs (*n* = 5) were sensitized subcutaneously and intraperitoneally with 0.5 mL per route (1 mL total) of a solution containing OVA (1 mg/mL; chicken egg albumin, grade II; Sigma, St. Louis, MO, USA) and aluminum hydroxide (1 mg/mL; J.T. Baker, Phillipsburg, NJ, USA) under controlled animal facility conditions. On day 8, animals received an antigenic boost via nebulized OVA (3 mg/mL) for 5 min using a Puritan-Bennett US-1 ultrasonic nebulizer (flow rate 2 mL/min, Shantou, GD, China.), connected to a Multistage Liquid Impinger (Burkard Manufacturing Co., Hertfordshire, UK). This system generated particles of <4 µm (44%), 4–10 µm (38%), and >10 µm (18%). After the procedure, guinea pigs were returned to the animal facility. The first antigenic challenge was administered 8 days after the boost using nebulized OVA (1 mg/mL, Shantou, GD, China.) following the same protocol. Additional challenges were performed on days 25 and 35 with nebulized OVA (0.5 mg/mL) (Figure 1A). Control animals (*n* = 5) received physiological saline under identical conditions. The acute bronchial obstructive response was evaluated using barometric plethysmography in freely moving animals, following Hamelmann et al. [22]. A whole-body single-chamber plethysmograph (Buxco Electronics Inc., Wilmington, NY, USA) measured pressure variations through a differential transducer connected to a preamplifier. Changes in pressure resulting from the heating and humidification of inhaled and exhaled air were recorded, avoiding invasive procedures. After 20 min of chamber adaptation, a baseline bronchial obstruction index (BI) was obtained before the antigenic challenge and compared with the post-challenge BI. Measurements were taken every 15 s, and the BI was calculated using the mean of the final 5 min of each session. BI was calculated using:BI = [(expiratory time (s) − relaxation time (s))/relaxation time (s)] × (peak expiratory flow (cmH_2_O)/peak inspiratory flow (cmH_2_O))

### 2.3. Bronchial Reactivity

Bronchial reactivity was assessed on day 35 of the asthma model. A cumulative dose–response curve to histamine (0.001–0.32 mg/mL; Sigma, St. Louis, MO, USA) was performed before the OVA challenge, following the previously published protocol [23]. Minor modifications were applied to adapt the procedure to our laboratory conditions, including the timing of the third OVA challenge. The bronchial BI was recorded for 5 min at each histamine dose. The first curve was used to determine the dose at which histamine induced a threefold increase in baseline BI. Once the BI returned to approximately 50% of baseline, the third OVA challenge was administered [24]. Three hours later, a second cumulative dose–response curve to histamine was performed following the same protocol. Animals were continuously monitored throughout the procedure to ensure their welfare and minimize distress.

### 2.4. Exclusion Criteria

Exclusion criteria included guinea pigs that did not exhibit airway obstruction or AHR in response to the challenge. However, no animals met these exclusion criteria.

### 2.5. Protein Identification

For protein identification, guinea pigs were administered deep general anesthesia using sodium pentobarbital (28 mg/kg, intraperitoneally; PiSa, Tlajomulco de Zúñiga, JAL, Mexico). The depth of anesthesia was meticulously verified by the absence of pedal and palpebral reflexes, lack of response to deep nociceptive stimuli, muscle relaxation, and no signs of consciousness. Once deep general anesthesia was confirmed, euthanasia was performed via terminal cardiac puncture, adhering to the Mexican Official Standards. This procedure was supervised by the Institutional Bioethics Committee and followed the guidelines set by the AVMA/CCAC. Although the study mainly targets the lower airways, such as bronchi and bronchioles, adequate tissue for the proteomic analysis of ASM was gathered by creating pools from the tracheal mucosa of six guinea pigs in the control group and another pool from six guinea pigs in the asthma model group. The tissues were separately homogenized in a phosphate buffer (6.7 mM K_2_HPO_4_, pH 7.4, with 0.04 M KCl and 1 mM MgCl_2_, Sigma, St. Louis, MO, USA) at 4 °C. Samples were sonicated three times at 30% amplitude for 30 s with 1-min ice intervals (Vibra-cell 75185; Sonics and Materials Inc., Newtown, CT, USA). The resulting suspension was desalted and precipitated at −20 °C using acetone, 10% trichloroacetic acid, and 20 mM dithiothreitol (DTT; Sigma, St. Louis, MO, USA). Cell pellets were resuspended in phosphate buffer at 4 °C, and protein concentrations were measured using the DC Protein Assay Kit (Bio-Rad, Hercules, CA, USA). Proteins (30 μg per lane) were separated through conventional electrophoresis on NuPAGE™ Bis-Tris 4–12% gels under reducing conditions (2.5% 2-mercaptoethanol), as described by Laemmli [25]. The Precision Plus Protein All Blue marker (Bio-Rad, Hercules, CA, USA) served as the molecular weight reference. Protein separation was performed using a commercial mini-gel system (Mini-Protean II) with a Power Pack 3000 at 80 V (Bio-Rad, Hercules, CA, USA) for stacking and 120 V for resolution. After electrophoresis, the protein bands were manually excised using a micropipette. Bands corresponding to MYPT1 (130 kDa), as well as additional bands at 120, 75, 50, and 37 kDa, were selected. While similar bands were reported in [26], the bands excised in our study showed distinct differences. The protein bands were washed with an ACN solution (50:50 *v*/*v*), reduced using 10 mM DTT, and alkylated with 100 mM iodoacetamide. The proteins were then digested with trypsin (Promega V528A, Madison, WI, USA) at 37 °C for 18 h. Peptides were extracted using an ACN:H_2_O acid mixture (50:45:5 *v*/*v*), concentrated (Eppendorf 5301, Hamburg, HH, Germany), and desalted using a ZipTip C18 column (Millipore, Saint Charles, MO, USA). The samples were deposited in triplicate onto an α-cyano matrix plate and analyzed using a MALDI TOF/TOF 4800 mass spectrometer. MS/MS spectra were then searched using the Paragon algorithm in Protein Pilot software (version 5.0.1; SCIEX, Framingham, MA, USA), employing a confidence threshold of 66%.

### 2.6. Antibody Reactivity

The antibodies utilized in this study were monoclonal and specifically targeted human proteins, as there are no commercial versions available for guinea pigs (*Cavia porcellus*). To confirm cross-reactivity and accurate detection, a comparative sequence analysis was conducted. Epitopes recognized by each antibody were identified, and their conservation between human and guinea pig sequences was evaluated using BLAST (Basic Local Alignment Search Tool, version 2.13.0; NCBI, Bethesda, MD, USA) alignments from the NCBI database. The MYPT1 monoclonal antibody recognizes amino acids 707–981, showing more than 96% identity between *Homo sapiens* (NP_001137357.1) and *Cavia porcellus* (XP_003475988.1). The anti-phospho-MYPT1 (Thr696) antibody targets the region around phosphorylated Thr696, which exhibited 100% identity, as did the anti-phospho-MYPT1 (Thr850) antibody (amino acids 848–856). For regulatory kinases, the rabbit ROCK1 monoclonal antibody was generated against a central region of the protein. Although precise sequence information from the manufacturer was unavailable, alignment showed more than 98% identity between human (NP_005397.1) and guinea pig (XP_003474160.2). Similarly, the anti-ROCK2 antibody [EPR7141(B)], recognizing the C-terminal region, also displayed more than 98% identity (human XP_054200614.1; guinea pig XP_063110367.1). Finally, the PPP1R3 antibody (C-8), targeting RG1 (amino acids 759–774), showed 100% identity between human (NP_002702.2) and guinea pig (XP_003475179.2). These findings confirm that the selected antibodies have a high degree of epitope conservation, supporting their suitability for reliable protein detection in guinea pigs.

### 2.7. Protein Expression by Immunohistochemistry in Paraffin

The section corresponding to the left lower lung lobe of each guinea pig (*n* = 5 per group) was collected following euthanasia. These sections were fixed in 10% buffered formalin and, after 24 h, dehydrated through a series of steps: distilled water (10 min), 96% ethanol (10 min), absolute ethanol (10 min), and xylene (10 min). The tissues were then embedded in paraffin, and 3–4 μm sections were cut using a microtome. Slides were deparaffinized in an incubator at 30 °C for 30 min and rehydrated in sequence with xylene (10 min), absolute ethanol (5 min), 96% ethanol (5 min), and distilled water (10 min). Antigen retrieval was performed in a microwave oven for 10 min using a 10 mM sodium citrate buffer (pH 6). To block non-specific binding sites, a 2% horse serum solution (Universal R.T.U. Vectastain Kit, Vector Laboratories; Burlingame, CA USA) was used. Samples were then incubated overnight at 4 °C with monoclonal antibodies: MYPT1 (1:25, Proteintech Group, Inc.; Rosemont, IL, USA), MYPT1-Thr696 (1:25, MyBioSource, San Diego, CA, USA), MYPT1-Thr850 (1:25, Sigma, USA), ROCK1 (1:25, MyBioSource, USA), ROCK2 (1:25, Abcam, Cambridge, UK), and PPP1R3A (RG1, 1:50, Santa Cruz Biotechnology, Dallas, TX, USA). Blank control slides underwent the same process but without the primary antibody. Both samples and controls were treated with 3% hydrogen peroxide (30% solution, Hycel S.A. de C.V., Zapopan, JAL, Mexico) and then incubated with the secondary antibody from the Universal R.T.U. Vectastain Kit for 1 h at room temperature. After washing, the slides were treated with a peroxidase/streptavidin complex from the same kit. Positive staining was revealed using the chromogenic substrate DAB (3,3′-Diaminobenzidine). Counterstaining was done with Mayer’s hematoxylin for 15 min. Slides were washed twice with 1% TBS-Tween after each step. Images were captured at 40× magnification using an optical microscope. ImageJ-Fiji (ImageJ 2.16.0/1.52g; Java 1.8.0_322) software was employed to quantify positive staining. Five random quadrants from bronchi smooth muscle images of each guinea pig in both control and asthma groups were selected. Color deconvolution was applied to separate DAB (chromogen) and hematoxylin (counterstain). The DAB image was converted to black-and-white pixels, with white pixels representing positive staining and black pixels representing background. A pixel histogram was generated to compare staining intensity, analyzing peaks corresponding to white (positive) and black (background) pixels.

### 2.8. Statistical Analysis

Normality and homogeneity of variance were checked before analysis. To assess differences in airway obstruction across antigenic challenges, a two-way ANOVA was used. Post hoc comparisons were made using Tukey’s test. Bronchial reactivity to histamine was measured by calculating the mean 200% provocative dose (PD200), which is the histamine dose that caused a three-fold increase in baseline bronchial BI. Changes in histamine responsiveness after antigenic challenge were expressed as the PD_200_ ratio, calculated as PD_200_ after OVA challenge divided by PD_200_ before the challenge. Significant differences between the control and allergic asthma groups were evaluated using an unpaired Student’s *t*-test. Relationships between pixel levels indicating protein expression and changes in airway reactivity, as well as correlations between protein expression levels, were analyzed using Pearson’s correlation coefficient. Statistical significance was defined as a two-tailed *p*-value less than 0.05. All analyses were performed using GraphPad Prism 6 (version 6.0).

## 3. Results

### 3.1. Allergic Asthma Model in Guinea Pigs

A temporary rise in the BI was detected in guinea pigs sensitized to the antigen after OVA challenge. The asthma model group exhibited a markedly higher peak BI following antigen exposure than the control group (*p* < 0.01; *n* = 5 per group; Figure 1B). Moreover, the post- to pre-antigen challenge PD200 ratio for histamine was significantly reduced in the asthma group compared with controls (*p* < 0.01; *n* = 5 per group; Figure 1C), demonstrating increased histamine responsiveness in the asthma model animals.

### 3.2. Protein Identification by Mass Spectrometry and Conventional Electrophoresis

Tracheal mucosa was dissected and subjected to analysis using mass spectrometry following separation by conventional electrophoresis (Figure 2A). Bands corresponding to the molecular weight of the MYPT1 protein, i.e., 130 kDa, as well as additional bands at molecular weights of 120, 75, 50, and 37 kDa, were selected (Figure 2B). These were associated with potential MYPT1 fragments reported in the literature [11,12,13]. Mass spectrometry analysis identified 16 proteins in the gels from both groups, based on the UniqueScore values obtained using the Paragon algorithm within the ProteinPilot (version 5.0.1) software, with a confidence level of 66%. Table 1 lists the proteins identified in the control and asthma model groups, highlighting the presence of the regulatory subunit of glycogen-targeted phosphatase 1 (RG1) in the guinea pig asthma model group.

### 3.3. Localization and Expression of Proteins in Bronchial Smooth Muscle

Since the proteomic analysis unexpectedly identified the presence of RG1, its evaluation was included alongside the assessment of other proteins closely associated with AHR. The localization and expression levels of MYPT1 and its phosphorylated forms at threonine residues 696 and 850 (MYPT1-Thr^696^ and MYPT1-Thr^850^), as well as ROCK1, ROCK2, and the regulatory subunit RG1, were assessed by immunohistochemistry using specific monoclonal antibodies (Figure 3A). All analyzed proteins were expressed in the bronchial smooth muscle, with a significant increase in the expression of RG1 and ROCK2 in guinea pigs from the asthma model compared to control guinea pigs (*p* < 0.01; *n* = 5 per group; Figure 3B). In Figure 4, the expression levels of RG1 and ROCK2 proteins showed a statistical significative inverse correlation with the PD200 ratio (*p* < 0.05; *n* = 5), suggesting that the increased expression of RG1 and ROCK2 is associated with enhanced AHR to histamine.

## 4. Discussion

The allergic asthma model in guinea pigs has been established as a valuable experimental tool for studying the pathophysiology of asthma, as it consistently reproduces the functional characteristics of the disease, including AHR and bronchial obstruction [27,28]. AHR, which has been shown to markedly affect small airways, has been the focus of numerous studies aimed at identifying the proteins involved in its development [29,30]. In this context, the present study demonstrated the identification of several proteins, among which RG1 was of particular interest, as it exhibited overexpression that showed a positive correlation with increased AHR in the bronchial smooth muscle of a guinea pig model of allergic asthma.

An important aspect of this study was the exclusive use of male guinea pigs, a decision based on previous observations indicating that some females did not develop AHR. This finding was decisive for their exclusion, as previous experimental asthma models—such as those conducted in rats—have demonstrated that sex hormones can influence protein expression in ASM, modulating both contractility and protein dynamics [31].

Given the previously documented importance of MYPT1 in modulating AHR, mass spectrometry—an established technique in proteomic analysis—was utilized to specifically identify this protein, both in its full-length form (approximately 130 kDa) and in potential fragments of 120, 75, 50, and 37 kDa reported in the literature [9,10,11,12]. Although neither MYPT1 nor its fragments were detected, the proteomic analysis identified several other proteins potentially involved in the regulation of ASM contractility, contributing to the protein expression profile associated with AHR that we had previously characterized [26]. These proteins included desmin, annexin A5, precursor protein FAM3A, tropomyosins 1 and 2, N-acetylgalactosaminyltransferase 18, type I and II collagens, SH3-domain-containing adaptor protein SPIN90, group IID secretory phospholipase A2, actin-binding LIM protein 2, ubiquitin E3 ligase NHLRC1, small U3 ribonucleoprotein, cytoplasmic malate dehydrogenase type 1, and notably, RG1.

RG1 is one of the regulatory subunits of the phosphatase PP1c, known for directing its activity toward the regulation of glycogen synthesis in smooth muscle [32,33]. Specifically, RG1 modulates the dephosphorylation of key enzymes such as glycogen synthase and glycogen phosphorylase, promoting intracellular glycogen accumulation [34]. Notably, several studies have reported functional competition and cross-regulation between the regulatory subunits MYPT1 and RG1 [35,36]. Both isoforms interact with the same high-affinity binding sites—RVxF and KIQF—that correspond to the general motif (R/K)(V/I)XF in PP1c [8], suggesting the possibility of direct competition for association with the phosphatase [37,38], and in rabbit portal vein, RG1 was observed to slow relaxation [35].

Among the molecular mechanisms described, the inhibition of MLCP has been recognized as a critical process in the regulation of smooth muscle contraction in AHR [3]. This process frequently involves phosphorylation-dependent inhibition of both PP1c—through endogenous inhibitors—and MYPT1 via direct phosphorylation [3,39]. Among the inhibitory mechanisms targeting MYPT1, post-translational modifications, particularly the phosphorylation of specific serine and threonine (Thr) residues, have been identified as key regulatory events. In humans, MYPT1 phosphorylation at Thr^696^ and Thr^850^ has been established as a major inhibitory mechanism that reduces MLCP activity, thereby promoting sustained smooth muscle contraction. However, the precise physiological relevance of these phosphorylation events remains controversial, as it has been proposed that they may contribute to the formation of non-functional phosphorylated isoforms. Alternatively, these modifications may play an important structural role by facilitating the transient dissociation of the holoenzyme complex and dynamically regulating its assembly and function in response to intracellular signaling cues [40,41].

The threonine phosphorylation events are mediated by several kinases that act through signaling cascades dependent on the small GTPase RhoA. Among the most relevant kinases are integrin-linked kinase (ILK), zipper-interacting protein kinase (ZIPK), and Rho-associated coiled-coil containing kinases (ROCK), which exist in two main isoforms: ROCK1 and ROCK2 [3,42]. Among these kinases, ROCK has been identified as the key enzyme responsible for phosphorylating MYPT1 at Thr696 and Thr850 in ASM. Its overactivation and the subsequent phosphorylation of MYPT1 have been strongly linked to prolonged contraction and heightened AHR in inflammatory conditions such as asthma [3,43]. Both ROCK isoforms are expressed in ASM; indeed, some studies have reported higher levels of ROCK1 expression in this tissue [44].

Since MYPT1 fragments were not identified in the study, the analysis was redirected toward exploring MLCP inhibition mediated by MYPT1 phosphorylation. It is important to highlight that the identification of RG1 via mass spectrometry was conducted on tracheal smooth muscle. This represents a limitation, as the trachea does not fully capture the functional and molecular characteristics of smooth muscle in the lower airways [45]. Indeed, previous studies from our group have demonstrated significant differences in the expression of several proteins between tracheal and bronchial smooth muscle, particularly in asthma models [26]. To obtain a more accurate and representative understanding of the phenomena associated with AHR, we evaluated the expression of RG1 and related proteins by immunohistochemistry in bronchial smooth muscle.

Given the established role of ROCK kinases in MYPT1 phosphorylation during the development of AHR, and the competitive interaction of RG1, the expression of ROCK1 and ROCK2 isoforms, as well as RG1 and MYPT1 together with their phosphorylated forms (MYPT1-Thr^696^ and MYPT1-Thr^850^), was evaluated. Interestingly, although immunohistochemical analysis revealed no changes in MYPT1 expression or its phosphorylated forms in the bronchial smooth muscle of guinea pig asthma model, both RG1 and ROCK2—but not ROCK1—showed significantly increased expression levels. This overexpression correlated positively with the degree of AHR. Moreover, a significant positive correlation was observed between ROCK2 and RG1, but not among the other proteins analyzed (Pearson correlation, *r* = 0.723, *p* = 0.018). These associations, together with the increase in ROCK2 and AHR, do not establish a causal relationship but rather provide descriptive evidence suggesting a potential functional involvement of RG1 in the regulation of ASM contraction associated with AHR. In this context, the identification of RG1 represents a relevant observational finding that may indicate a novel signaling pathway modulating AHR through a competitive interaction with MYPT1. However, further functional studies are required to confirm this hypothesis and elucidate the underlying mechanisms.

This study has certain limitations, including the lack of species-specific antibodies, the absence of validation in human samples, and the lack of functional or mechanistic assays directly linking RG1 to contractile responses. Additionally, although the guinea pig model is widely used to investigate AHR, it carries inherent constraints associated with animal models, including potential differences in the molecular regulation of ASM compared with humans. Some of these limitations are intrinsic to the descriptive and exploratory nature of the study. Nevertheless, these factors do not diminish the robustness of the findings. The identification of RG1 as a protein associated with AHR for the first time, along with the overexpression of ROCK2, represents a pioneering finding that expands our understanding of contractile regulation in asthma and provides a foundation for future investigations of proteins potentially involved in AHR. Furthermore, the absence of changes in MYPT1 phosphorylation at Thr696 and Thr850 suggests that ROCK2-mediated inhibition may occur through alternative or non-canonical pathways, such as phosphorylation at Ser507 previously reported in asthma models, guiding future studies toward alternative phosphorylation sites [14]. Moreover, given the high degree of conservation of the RG1 and ROCK2 signaling pathways between rodents and humans, these findings may be extrapolated to human airway physiology, underscoring their potential relevance to the pathophysiology of asthma.

Overall, the findings of this study do not establish causality, but they highlight the relevance of RG1 as a potential regulator of AHR in ASM. The observed overexpression of RG1 in the allergic asthma guinea pig model is positively associated with AHR and with ROCK2, suggesting a possible functional interaction with the MLCP pathway, primarily with MYPT1, which could influence the modulation of smooth muscle contractility. These results provide a descriptive framework for future studies aimed at investigating the functional mechanisms of RG1 in asthma pathophysiology and its potential as an emerging therapeutic target associated with AHR.

## 5. Conclusions

In conclusion, this descriptive and exploratory study identifies, for the first time, an association between the RG1 protein and AHR in a guinea pig model of allergic asthma, alongside the concomitant overexpression of ROCK2. Although these findings do not establish a direct functional relationship, they highlight RG1 as a potential key regulator of bronchial smooth muscle contraction, likely through competitive interactions with MYPT1. Additionally, the proteomic characterization revealed other proteins differentially expressed in ASM that, like RG1, may play relevant roles in modulating AHR. Collectively, these results expand the molecular landscape of AHR and provide a robust framework for future mechanistic studies focused on RG1 and the exploration of novel regulatory proteins as potential targets in asthma.

## Figures and Tables

**Figure 1 biomedicines-13-03111-f001:**
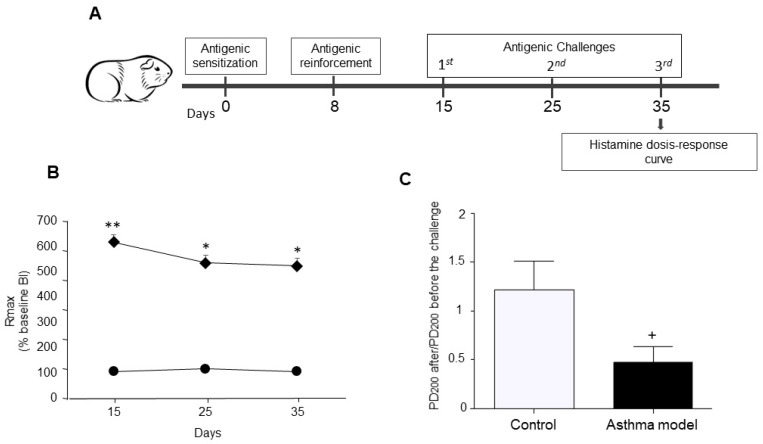
Experimental design. (**A**) Allergic asthma model in guinea pigs. (**B**) Maximum broncho-obstructive index (BI) recorded after antigenic challenge in OVA-sensitized guinea pigs (diamonds) compared to control animals receiving saline (circles). (**C**) Evaluation of airway hyperresponsiveness to histamine: Difference in the mean provocative dose required to reach a 200% increase in BI before and after antigen exposure, in response to increasing histamine doses. The PD_200_ ratio was calculated as the PD_200_ value observed after antigen challenge divided by the PD_200_ value measured prior to the challenge. *n* = 5 per group; * *p* < 0.05, ** *p* < 0.01, two-way ANOVA using Tukey’s post-hoc test. + *p* < 0.01, unpaired Student’s *t*-test.

**Figure 2 biomedicines-13-03111-f002:**
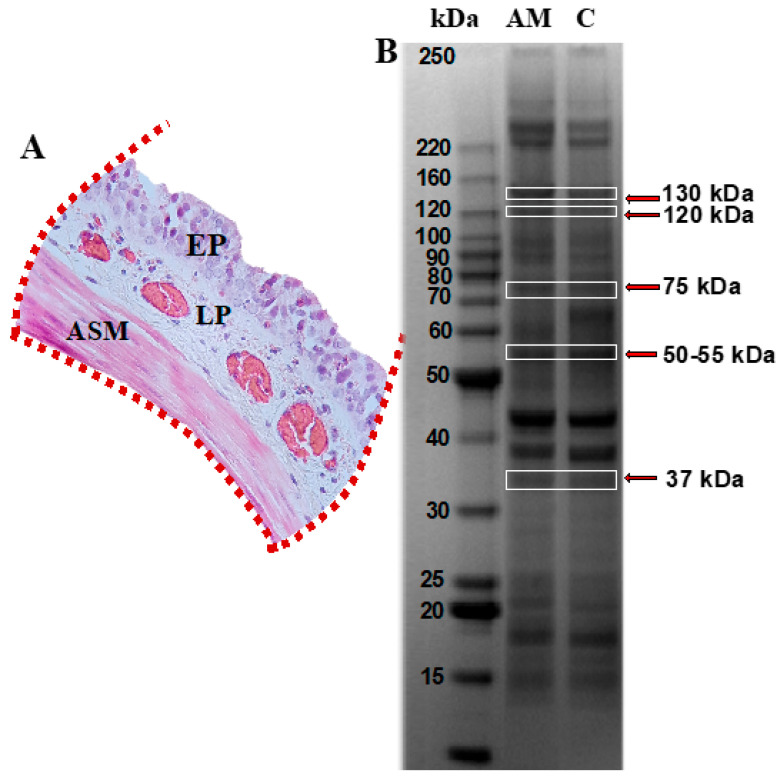
Protein profiling in tracheal smooth muscle from guinea pigs within the asthma model. (**A**) Tracheal segment isolated and sectioned, highlighting the epithelium (EP), lamina propria (LP), and airway smooth muscle (ASM) regions, indicated with red dotted lines. (**B**) SDS-PAGE polyacrylamide gel. Lane 1 corresponds to the asthma model group (AM), and lane 2 to the control group; *n* = 6 guinea pigs per group. Protein bands identified with red arrows, corresponding to molecular weights of approximately 130, 120, 75, 50–55, and 37 kDa, were excised and analyzed by mass spectrometry using the MALDI-TOF-TOF MS/MS system. Image captured using the ChemiDoc Imaging System (Bio-Rad).

**Figure 3 biomedicines-13-03111-f003:**
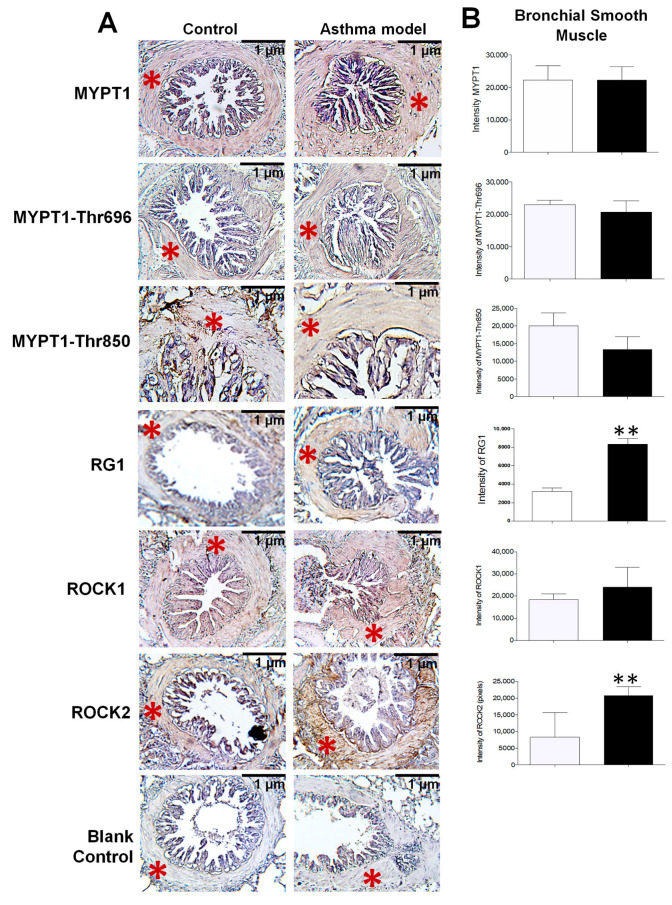
Expression and distribution of proteins in the bronchial smooth muscle of guinea pigs from the asthma model. (**A**) Immunohistochemistry image illustrating the localization of positively stained proteins in the bronchial smooth muscle of guinea pigs from both the asthma model and control groups (red asterisk). The blank control represents the staining control. Chromogen: DAB; counterstained with hematoxylin. Optical microscope, 40× magnification. (**B**) Relative intensity of positive staining for the regulatory subunit known as myosin phosphatase target subunit 1 (MYPT1) and its phosphorylated forms at Thr^650^ (MYPT1-Thr^650^) and Thr^850^ (MYPT1-Thr^850^), isoforms of Rho-associated kinases (ROCK1 and ROCK2), and the glycogen-targeting regulatory subunit 1 of protein phosphatase 1 (RG1), based on pixel intensity. White bars correspond to the control group and black bars to the asthma model group. Analysis performed using ImageJ-Fiji software, *n* = 5 guinea pigs per group, unpaired Student’s *t*-test, ** *p* < 0.01.

**Figure 4 biomedicines-13-03111-f004:**
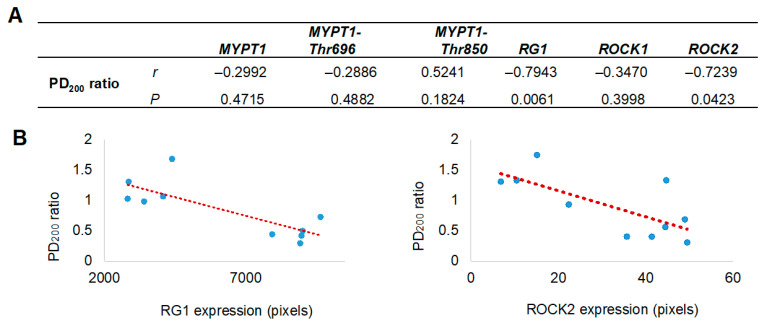
Correlation between airway hyperresponsiveness and the levels of selected proteins in bronchial smooth muscle. (**A**) Pearson correlation coefficients (*r*) and probability values (*p*) between protein expression in bronchial smooth muscle and airway hyperresponsiveness values (PD_200_ ratio) are presented. (**B**) Scatter plots illustrate a positive correlation between the expression levels of Rho-associated kinase isoform 2 (ROCK2) and the glycogen-targeting regulatory subunit 1 of protein phosphatase 1 (RG1) with the PD200_200_ ratio, which represents the change in the histamine dose required to induce airway obstruction after antigenic challenge, compared to the baseline (pre-challenge) value. The red dashed line represents the fitted regression trend. MYPT1 refers to the myosin phosphatase target subunit 1, including its phosphorylated forms at Thr^650^ (MYPT1-Thr^650^) and Thr^850^ (MYPT1-Thr^850^).

**Table 1 biomedicines-13-03111-t001:** Proteins identified in tracheal tissue from guinea pigs in the asthma model and control group.

Molecular Weight	Control	Asthma Model
130 kDa	Collagen α-1 (COLA1)	Collagen α-1 (COLA1)
	Collagen α-2 (COLA2)	
120 kDa	E3 ubiquitin-protein ligase (NHLRC1)	Polypeptide N-acetylgalactosaminyltransferase 18 (GALNT18)
	U3 small nucleolar ribonucleoprotein (IMP)	
75 kDa	Collagen α-1 (COLA1)	Collagen α-1 (COLA1)
		Glycogen-targeted protein phosphatase type 1 regulatory subunit (RG1)
		FAM3A protein precursor (FAM3A)
		SH3 adapter protein (SPIN90)
		Secretory phospholipase A2 Group IID (PLAIIG4A)
		Actin-binding LIM protein 2 (ABLIM2)
50 kDa		Desmin (DES)
37 kDa	Annexin A5 (ANXA5)	Annexin A5 (ANXA5)
	Malate dehydrogenase, cytoplasmic (MDH1)	Tropomyosin α-1 (TPM1)
		Tropomyosin β (TPM2)

## Data Availability

The original contributions presented in this study are included in the article. Further inquiries can be directed to the corresponding authors.

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
