# Peer review of "Potential Role of the Glycogen-Targeting Phosphatase Regulatory Subunit in Airway Hyperresponsiveness in Asthma"

_biomedicines, 2025, doi:10.3390/biomedicines13123111_

Round 1

Reviewer 1 Report

Comments and Suggestions for Authors

The manuscript by Alvarez-González et al. presents a descriptive and exploratory study investigating protein expression profiles in airway smooth muscle (ASM) using a guinea pig model of allergic asthma. The authors identify the regulatory subunit of glycogen-targeted phosphatase 1 (RG1) as a novel protein associated with airway hyperresponsiveness (AHR), alongside Rho-associated kinase 2 (ROCK2). The study is well-structured, methodologically sound, and addresses an important gap in our understanding of AHR mechanisms. Thus, the manuscript contains a set of novel valuable data, is well-written, clearly illustrated, and easy to follow. Nevertheless, I have a few suggestions that I believe could further improve the presentation.

- The rationale for selecting only a specific subset of protein bands for subsequent mass spectrometry analysis is unclear. The authors should briefly explain why other prominent bands were not analyzed, which would help clarify the comprehensiveness of their proteomic approach.

- The inclusion of a schematic for the experimental design (Fig. 1A) is very helpful. However, the figure legend should describe the design in more detail, specifically explaining the meaning of the annotations "1°", "2°", and "3°".

Author Response

Query 1. The rationale for selecting only a specific subset of protein bands for subsequent mass spectrometry analysis is unclear. The authors should briefly explain why other prominent bands were not analyzed, which would help clarify the comprehensiveness of their proteomic approach.

Reply 1. We greatly appreciate your comment as it helps to clarify the objective of our work. As stated in the Introduction, MYPT1, the protein we aimed to identify, has an approximate molecular weight of 130 kDa. However, since it was not observed, we investigated bands where functional fragments documented with smaller molecular weights (120, 75, 50, and 37 kDa) could be found. Nevertheless, mass spectrometry analysis did not detect these potential proteolytic fragments. The remaining bands were analyzed as well, though without success, and they were part of another study on actins and other proteins recently published (1). To clarify these points, we included the Protein Identification section in Materials and Methods, detailing which bands were selected (lines 180-183). Additionally, we added text in the Discussion to clarify that other bands characterized in the other study (1) also did not show the expression of this protein (lines 399-411).

  1. Álvarez-González M, Flores-Flores A, Carbajal-Salinas V, Bazán-Perkins B. Altered actin isoforms expression and enhanced airway responsiveness in asthma: the crucial role of β-cytoplasmic actin. Front Physiol. 2025;16:1627443. doi:10.3389/fphys.2025.1627443.

Query 2. The inclusion of a schematic for the experimental design (Fig. 1A) is very helpful. However, the figure legend should describe the design in more detail, specifically explaining the meaning of the annotations "1°", "2°", and "3°".

Reply 2. We appreciate the observation regarding Figure 1A of the manuscript. We have reorganized the schematic in order in the figure to enhance its clarity, structure, and overall comprehensibility for the reader.

We appreciate the review provided for our manuscript. Your insights have been invaluable in improving the quality of our work.

Reviewer 2 Report

Comments and Suggestions for Authors
  1. Abstract : It has no structure, ie. Introduction, aim/objective, method and material, results, conclusions
  2. Clarify the Rationale and Hypothesis
  • Explain why MYPT1 was initially chosen as a target and what specific evidence in the literature suggests its involvement in AHR.
  • Provide a clearer rationale for investigating RG1, since it appears unexpectedly in the proteomic screen.

Address Gaps and Limitations

  • Explicitly state the limitations of using tracheal tissue instead of lower airway smooth muscle.
  • Acknowledge constraints of animal models
  • Note the lack of functional assays directly linking RG1 to contractile responses.

The number of animals was determined based on their availability…IT IS NEEDED A Justification based on power estimation, previous studies, or 3Rs, but not availability.

Mandatory corrections for scientific and ethical accuracy:

  • Do not justify sample size by availability.
  • Replace “arbitrarily” with “randomly.”
  • Correct OVA concentration inconsistencies.
  • Fix “US-1 Benneô€„´ nebulizer” to proper naming.
  • Clarify modifications to the protocol.
  • Ensure timeline is consistent and easy to follow.
  • Improve flow and reduce repetition.
  • Consider reorganizing into shorter sentences.

clarify why tracheal tissue is used to represent lower airway smooth muscle.

The reported pentobarbital dose (28 mg/kg) :

  • Verify the dose, or
  • Provide a citation demonstrating that 28 mg/kg is lethal in guinea pigs under the conditions used.

What is the value of the study

Are there any bias ?

Comments on the Quality of English Language

NEED IMPROVEMENT

Author Response

Query 1. Abstract: It has no structure, ie. Introduction, aim/objective, method and material, results, conclusions.

Reply 1. Thank you for your comment. We have restructured the abstract in order to present a clearer and more coherent organization of the elements described in the study.

Query 2. Clarify the Rationale and Hypothesis.

Reply 2. We appreciate the observation regarding the lack of clarity in this section. The rationale and hypothesis have now been reinforced and more clearly articulated in lines 78-93.

Query 3. Explain why MYPT1 was initially chosen as a target and what specific evidence in the literature suggests its involvement in AHR.

Reply 3. We have articulated the importance of MYPT1 in airway hyperresponsiveness in lines 78-89. We appreciate this insightful comment; we believe that this modification has enhanced the clarity of the initial objective of our study.

We chose not to elaborate further on the kinase-mediated signaling and regulatory pathways of MYPT1 in this section, as we consider this to be an important aspect that is more appropriately and clearly addressed in the Discussion section (lines 439-443).

Query 4. Provide a clearer rationale for investigating RG1, since it appears unexpectedly in the proteomic screen.

Reply 4. Indeed, RG1 emerged unexpectedly in our proteomic analysis, as it was not among the proteins initially considered within our MYPT1-centered hypothesis. For this reason, its inclusion could not be justified on the basis of prior theoretical assumptions within our original conceptual framework. Nevertheless, we added a brief explanation in the Results section (Lines 304-305) indicating why this finding was relevant for its subsequent evaluation, without delving into regulatory mechanisms, which are more appropriately and comprehensively addressed in the Discussion section. In this way, we maintain the narrative coherence of the manuscript while acknowledging that RG1 was incorporated as a result of its proteomic identification rather than through an a priori hypothesis.

Query 5. Explicitly state the limitations of using tracheal tissue instead of lower airway smooth muscle.

Reply 5. We fully agree that the use of tracheal smooth muscle entails important limitations, as in a previous study (1), we observed that tracheal smooth muscle may exhibit significant variations in protein expression patterns compared to the lower airways. Nevertheless, the mass spectrometry study required a large amount of tissue, which is why we initially used tracheas. These limitations, as well as the rationale for redirecting the analysis toward bronchial smooth muscle, have been incorporated into the manuscript in lines 447-455.

  1. Álvarez-González M, Flores-Flores A, Carbajal-Salinas V, Bazán-Perkins B. Altered actin isoforms expression and enhanced airway responsiveness in asthma: the crucial role of β-cytoplasmic actin. Front Physiol. 2025;16:1627443. doi:10.3389/fphys.2025.1627443.

Query 6. Acknowledge constraints of animal models. Note the lack of functional assays directly linking RG1 to contractile responses.

Reply 6. We fully agree with the need to explicitly state the limitations associated with the absence of mechanistic assays directly linking RG1 to the regulation or contractile mechanisms of airway smooth muscle that contribute to AHR. The only evidence we have is described in the article by Gailly (1), where in rabbit portal vein RG1 slowed the relaxation, suggesting that it competes with MYPT1 in PP1c, and the study by chicken gizzard (2), which confirms that MYPT1 competes with RG1 for the regulatory site in PP1c. Accordingly, we have incorporated these considerations into the manuscript, which can be found in lines 412-420.

  1. Gailly, P., Wu, X., Haystead, T. A., Somlyo, A. P., Cohen, P. T., Cohen, P., & Somlyo, A. V. (1996). Regions of the 110-kDa regulatory subunit M110 required for regulation of myosin-light-chain-phosphatase activity in smooth muscle. European journal of biochemistry, 239(2), 326–332. https://doi.org/10.1111/j.1432-1033.1996.0326u.x
  2. Johnson, D. F., Moorhead, G., Caudwell, F. B., Cohen, P., Chen, Y. H., Chen, M. X., & Cohen, P. T. (1996). Identification of protein-phosphatase-1-binding domains on the glycogen and myofibrillar targetting subunits. European journal of biochemistry, 239(2), 317–325. https://doi.org/10.1111/j.1432-1033.1996.0317u.x

Query 8. The number of animals was determined based on their availability…IT IS NEEDED A Justification based on power estimation, previous studies, or 3Rs, but not availability.

Do not justify sample size by availability.

Reply 8. In our study, the number of guinea pigs was primarily determined according to the principles of the 3Rs, ensuring the use of the minimum number of animals necessary to obtain reproducible and statistically interpretable results without compromising their welfare. Moreover, previous studies from our group have demonstrated that this asthma model exhibits highly consistent physiological and molecular responses across animals, supporting the robustness of the model and allowing the use of reduced sample sizes without affecting the reliability of the findings (1). We have revised this section of the manuscript accordingly and added the appropriate justification for the sample size, which can be found in lines 110-112.

  1. Álvarez-González, M., Pacheco-Alba, I., Moreno-Álvarez, P., Rogel-Velasco, L., Guerrero-Clorio, S., Flores-Flores, A., ... & Bazán-Perkins, B. (2025). Phenotypes of antigen-induced responses in guinea pigs: Beyond the asthma model. Molecular Immunology, 179, 1-8.

Query 9. Replace “arbitrarily” with “randomly.”

Reply 9. We have corrected the term “arbitrarily” and replaced it with “randomly” in line 113.

Query 10. Correct OVA concentration inconsistencies.

Reply 10. Thank you for your observation. We have updated the ovalbumin concentration used during allergic sensitization to 1 mg/ml. This modification can be found in line 120.

Query 11. Fix “US-1 Benne nebulizer” to proper naming.

Reply 11. The correct name of the nebulizer has been updated in line 123.

Query 12. Clarify modifications to the protocol.

Reply 12. The adjustments made to the protocol described by Campos et al. (2001) have been clearly detailed in lines 115-122 of the manuscript.

Query 13. Ensure timeline is consistent and easy to follow.

Reply 13. We have carefully revised the manuscript to ensure that the experimental timeline is consistent, clear, and easy to follow.

Query 14. Improve flow and reduce repetition.

Reply 14. We have thoroughly revised the manuscript to improve the overall flow and readability. Redundant statements have been removed, and complex sentences have been restructured into shorter, clearer sentences to enhance clarity and logical progression throughout the text.

Query 15. Consider reorganizing into shorter sentences.

Reply 15. We have revised the manuscript to reorganize longer, complex sentences into shorter, more concise sentences. This restructuring improves readability and ensures that each statement conveys a single clear idea, facilitating comprehension.

Query 16. The reported pentobarbital dose (28 mg/kg) :

  • Verify the dose, or
  • Provide a citation demonstrating that 28 mg/kg is lethal in guinea pigs under the conditions used.

Reply 16. We fully agree that the pentobarbital sodium dose used (28 mg/kg, intraperitoneal) requires detailed clarification. In our study, this dose was not used as a standalone method of euthanasia, but rather to induce deep general anesthesia, which was subsequently completed with an internationally accepted adjunctive method—terminal cardiac puncture. This approach is consistent with the principles of the 3Rs and with international guidelines (AVMA Guidelines for the Euthanasia of Animals, 2020; CCAC Guidelines on the Euthanasia of Animals Used in Science), which allow terminal invasive procedures provided that the animal is under clinically verified deep general anesthesia, even when the administered dose is not lethal by itself.

Although Tabari and Becker (1958) reported anesthetic effects of pentobarbital in guinea pigs, we acknowledge that available evidence does not conclusively demonstrate that 28 mg/kg (nor any fixed dose, as lethality depends on model- and condition-specific factors) administered intraperitoneally is consistently lethal. Therefore, in our protocol we did not assume pharmacological lethality based solely on the animals’ apparent unresponsiveness. Instead, we rigorously verified deep anesthesia by confirming absence of the pedal reflex, palpebral reflex, and response to deep nociceptive stimuli, as well as by assessing muscle tone, righting reflex, and any sign of consciousness. Only after confirming deep general anesthesia, and in accordance with Mexico normativity (NOM-062-ZOO-1999, NOM-033-SAG/ZOO-2014), and AVMA/CCAC guidelines, was terminal cardiac puncture performed as an adjunctive method, ethically and appropriately completing the euthanasia process.

We believe this clarification adequately addresses the reviewer’s concern, and it has been incorporated into lines 159-165 of the revised manuscript.

Query 17. What is the value of the study? Are there any bias?

The value of the study lies in identifying new protein markers, i.e. RG1 and ROCK2, that may contribute to airway hyperresponsiveness in asthma, which could lead to improved understanding and potential therapeutic targets. The study acknowledges that bias could arise from the use of an animal model (guinea pigs) which may not fully replicate human asthma.

 We appreciate the review provided for our manuscript. Your insights have been invaluable in improving the quality of our work.

Round 2

Reviewer 2 Report

Comments and Suggestions for Authors

no more !

Comments on the Quality of English Language

no more comments